# Anti-Phototoxicity Effect of Phenolic Compounds from Acetone Extract of *Entada phaseoloides* Leaves via Activation of COX-2 and iNOS in Human Epidermal Keratinocytes

**DOI:** 10.3390/molecules27020440

**Published:** 2022-01-10

**Authors:** Yanisa Mittraphab, Yhiya Amen, Maki Nagata, Masako Matsumoto, Dongmei Wang, Kuniyoshi Shimizu

**Affiliations:** 1Division of Systematic Forest and Forest Products Sciences, Department of Agro-Environmental Sciences, Graduate School of Bioenvironmental Sciences, Kyushu University, Fukuoka 819-0395, Japan; yanisa.mitt@gmail.com (Y.M.); yhiyaamen@gmail.com (Y.A.); makinagata0817@gmail.com (M.N.); mandm8010@hotmail.com (M.M.); spswdm@hotmail.com (D.W.); 2Department of Pharmacognosy, Faculty of Pharmacy, Mansoura University, Mansoura 35516, Egypt

**Keywords:** *Entada phaseoloides*, photoprotection, UVB, HaCaT cells, cosmeceuticals

## Abstract

The extract from *Entada phaseoloides* was employed as active ingredients of natural origin into cosmetic products, while the components analysis was barely reported. Using LC-DAD-MS/qTOF analysis, eleven compounds (**1**–**11**) were proposed or identified from acetone extract of *E. phaseoloides* leaves (AE). Among them, six phenolic compounds, protocatechuic acid (**2**), 4-hydroxybenzoic acid (**3**), luteolin-7-*O*-β-d-glucoside (**5**), cirsimaritin (**6**), dihydrokaempferol (**9**), and apigenin (**10**), were isolated by various chromatographic techniques. Protocatechuic acid (**2**), epicatechin (**4**), and kaempferol (**11**) at a concentration 100 μM increased the HaCaT cells viability of the UVB-irradiated cell without any cytotoxicity effect and reduced the expression of COX-2 and iNOS inflammation gene. Moreover, compounds **2** and **4** could have potent effects on cell migration during wound closure. These results suggest that compounds **2**, **4**, and **11** from AE have anti-photoaging properties and could be employed in pharmaceutical and cosmeceutical products.

## 1. Introduction

*Entada phaseoloides* (Family: Fabaceae) is a huge evergreen that grows high in the tropical forest and is found in Africa, Asia, and Australia’s lowland coastal forests. The pods become large in length and width (2 m and 130 mm, respectively). The brown seeds (around 10–20 seeds) are contained inside, with a diameter size of approximately up to 100. In Southeast Asia and tropical regions, *E. phaseoloides* is a well-known traditional medicinal plant. It has been claimed that *E. phaseoloides* is used as folk therapy for a variety of diseases and has a wide range of therapeutic characteristics. People in tropical and sub-tropical regions use almost all portions of this plant in traditional medicine to treat a wide range of diseases, including hemorrhoids, stomachaches, toothaches, spasms, gastritis, and lymphadenitis [1]. The components and biological activities (such as anti-inflammatory activity and antioxidant) of the seeds and stems of *E. phaseoloides* were investigated. In 1989, the first report, *E. phaseoloides* leaves yielded entadamide C, a Sulphur-containing amide that was isolated and described [2]. Few data are available on compounds isolated from *E. phaseoloides* leaves. The isolated compounds from *E. phaseoloides* seeds and stems such as quercetin, luteolin, apigenin, and dihydrokaempferol have been reported [3]. Flavonoids have been linked to a wide range of biological activities, including illness prevention, therapy, and prevention of coronary heart disease, neurological diseases, and more [4]. Entadine^®^ is a photo-aging agent that shields the skin from both sun and indoor radiation. It is a skin-care ingredient derived from the seeds and stems of *E. phaseoloides*.

The human skin is the body’s most exposed organ and defends against a variety of environmental ailments. Meanwhile, chronic, and recurrent environmental harm to human skin is a real possibility. UV radiation from the sun is a type of high-energy electromagnetic radiation that is regarded to be phototoxic to all organisms. UV radiation can be divided in three types based on the wavelengths: UVA (315–400 nm), UVB (280–315 nm), and UVC (200–280 nm). The ozone layer in the atmosphere absorbs UVC. UVA and UVB radiation reach the ground in about 95% and 5% of cases, respectively [5]. UV light induces programmed cell death, or apoptosis, which is an important preventive mechanism for the skin against neoplastic transformation [6]. The UVB, a large part of the solar UV, is ineffective in penetrating into the deep skin layer; however, it particularly influences the epidermis (the skin’s superficial layer which mostly comprises of keratinocytes) [7]. The reactive oxygen species (ROS), such as peroxyl radicals, hydroxyl radicals, active precursors so-called singlet oxygen and superoxide radicals, such ROS could attack cellular organelles as well as cells membrane causing not only the damage in organelle but also lipid peroxidation. There are three consequences of UVB irradiation of HaCaT. Firstly, the growth of post-irradiation-dependent in ROS, secondly, decline of subsequent cell membrane fluidity, and lastly, the mitochondrial membrane depolarization. Moreover, solar radiation is one of the important factors potentially causing skin diseases and specialized epithelium [8]. Consequently, they decrease intracellular ROS levels that might be the powerful strategy for the protection of skin damage. Over the past years, the development of natural compounds and flavonoids (can be generally found in fruits, vegetables, green tea, red wine, and a variety of biological activities like antioxidation) have significantly drawn attention as preventive agents for fighting UVB-induced skin damage via scavenging ROS [9,10].

Our investigation: the protective effect of acetone extract from *E. phaseoloides* leaves (AE) on UVB-irradiated HaCaT cells. However, the active compounds from AE are not unknown. Currently, we isolated the components and investigated the protective effect of UVB-irradiated on HaCaT cells.

## 2. Material and Methods

### 2.1. Chemicals and Reagents

The solvents, *n*-hexane, dichloromethane (DCM), ethyl acetate (EtOAc), acetone, and methanol (MeOH), were purchased from Wako Pure Chemical Industries (Osaka, Japan). The standard compounds, epicatechin, gallic acid, and 4-hydroxybenzoic acid, were purchased from Wako Pure Chemical Industries (Osaka, Japan). Quercetin and luteolin were purchased from Sigma-Aldrich (St. Louis, MO, USA). Apigenin, protocatechuic acid, and kaempferol were purchased from Tokyo Chemical Industry (Tokyo, Japan).

### 2.2. Identification of the Isolated Compounds

^1^H and ^13^C-NMR spectra were recorded using a Bruker DRX-600 spectrometer (Bruker Daltonics, Billerica MA, USA); methanol-*d*_4_ (CD_3_OD-*d*_4_) and chloroform-*d* (CDCl_3_-*d*) were purchased from Cambridge Isotope Laboratories (Andover, MA, USA). Analytical TLC was carried out on silica gel 60 F_254_ plate (Merck, Darmstadt, Germany) using H_2_SO_4_ in methanol, followed by heating. Preparative TLC was performed on silica gel (Merck, Darmstadt, Germany).

### 2.3. Extraction and Isolation

Leaves of *E. phaseoloides* were collected from Kanchanaburi Province, Thailand, in June 2019 and authenticated by Ratanachai Suthima, a staff of Forestry Center in Kanchanaburi. The sample was dried under the shade for four days and then ground into a fine powder. The powdered *E. phaseoloides* leaves (200 g) was extracted by sonicator bath at 50 °C with acetone four times (4 L, 60 min each). Acetone extract was evaporated under reduced pressure, using a rotary vacuum evaporator at 45 °C to give acetone crude extract (9.529 g). Acetone crude extract was analyzed by HPLC and chromatogram. The acetone extract (9.592 g) was chromatographed over a silica gel (Wakogel C 200, pore size 7 nm, particle diameter 75–15 μm) column with gradient EtOAc: *n*-hexane mixture to afford 27 subfractions (EPL1-EPL27). Fraction EPL21 was purified again by preparative TLC (acetone-*n*-hexane, 3:7 *v*/*v*) to give **9** (0.62 mg). Fraction EPL22 was fractionated by silica gel column chromatography eluted with acetone-*n*-hexane (1:9 to 1:0 *v*/*v*) mixture to obtained **3** (6.01 mg). Fraction EPL23 was purified by chromatographic techniques to afford **3** (2.97 mg), **9** (7.61 mg), **6** (0.55 mg), and **10** (1.32 mg). Fraction EPL24 was isolated by chromatographic techniques to give **2** (12.11 mg). Fraction EPL 27 was fractionated by normal phase MPLC (BUCHI, Reverelis Prep Purification System, Switzerland), then subjected to reverse phase MPLC system to afford **5** (6.88 mg).

### 2.4. High-Performance Liquid Chromatography (HPLC) Analysis

All experiments were performed on a 1260 infinity II LC (Agilent Technologies, Santa Clara, CA, USA). The isolated compounds and extract were dissolved in methanol at the concentration of 1 and 3 mg/mL, respectively. They were filtered through Millipore 0.20 μm filters before injection for chromatographic separation (Millex-LG, Tokyo, Japan). Then, 5 μL of each was injected and the flow rate was 0.4 mL/min. The separation was performed on YMC triart-C18 column (4.6 × 150 mm, 5 μm), (YMC Company, Kyoto, Japan) at 40 °C with the gradient system of 0.1% formic acid in water (solvent A) and 0.1% formic acid in methanol (solvent B) as follows 0–2 min, 20% B; 2–22 min, 20–100% B; 22–30 min, 100% B, finally the B content was reduced to the initial conditions in 5 min and the column was re-equilibrated for 5 min. The diode array detector (DAD) was measured over the rage of 200–600 nm. The evaporation temperature of the ELSD detector was set at 80 °C. The nebulizer temperature of the ELSD detector was set at 30 °C.

### 2.5. Analysis of the Acetone Extract by LC-DAD-MS/qTOF

The powered *E. phaseoloides* leaves (100 mg) was extracted by sonication with 15 mL of acetone for 60 min at 50 °C. The extracted solution was filtered and then evaporated under reduced pressure. The acetone extract was prepared in methanol to concentration 500 ppm, it was filtered twice through Millipore 0.20 μm PTFE filters (Millex-LG, Tokyo, Japan). The experiments were performed on an Agilent 1290 Series UPLC system equipped with a 1290 photodiode array detector (DAD) (Agilent Technologies, Santa Clara, CA, USA) coupled to an Agilent 6545 q-TOF hybrid mass spectrometer (MS) with a dual electrospray ionization (ESI) source for simultaneous spraying of a mass reference solution that enabled continuous calibration of detected *m*/*z* values was used for the analysis of the samples. The column was a Poroshell 120 EC-C18 (2.1 × 100 mm, 2.7 μm), Agilent Technologies, Santa Clara, CA, USA). The gradient system of 0.1% formic acid in water (solvent A) and 0.1% formic acid in acetonitrile (solvent B) as follows 0–20 min, 10–95% B; 20–25 min, 95% B. The time for re-equilibration to initial conditions was 5 min.

### 2.6. Cell Culture

Human spontaneously transformed keratinocytes HaCaT cells were cultured in Dulbecco’s Modified Eagle’s medium High Glucose (DMEM, purchased from Wako, Osaka, Japan). The 10% heat-inactivated fetal bovine serum (FBS, purchased from Wako, Osaka, Japan) and antibiotics 100 U/mL penicillin-streptomycin (Wako, Osaka, Japan) were added to the medium. The cells were grown at 37 °C in 5% CO_2_ humidified incubator.

### 2.7. Cell Viability Measurement by MTT Assay

HaCaT cells were seeded into 96-well plate at a density 1 × 10^5^ cells/mL and were incubated overnight at 37 °C in humidified atmosphere of 5% CO_2_. Then, the cells were treated with isolated compounds at various concentrations for 24 h. Subsequently, 10 μL of 3-(4,5-dimethylthiazol-2-yl)-2,5-diphenyltetrazolium bromide (MTT, purchased from Tokyo Chemical Industry, Tokyo, Japan) solution (5 mg/mL in phosphate buffered saline, PBS) was added to each well and incubated for additional 4 h. Then, supernatant was discarded, and 40 mM HCl-isopropanol (100 μL) was added to dissolve formazan crystals. The absorbance was measured at 570 nm using microplate reader. The percentage of cell viability measured in control cells treated with DMSO without samples was used to calculate cell viability.

### 2.8. UVB-Irradiation

HaCaT cells were seeded into a 96-well plate for 24 h and then pre-treatment with the testing compounds at various concentrations for 24 h at 37 °C in 5% CO_2_ humidified incubator. The cells were washed with phosphate buffered saline (PBS) one time and added 100 μL of PBS. After that, UVB-irradiation was exposed to HaCaT cells (30, 50, and 100 mJ/cm^2^). UVB-irradiation was performed with Bio-Link Crosslinker (Vilber Loubert Biolink^TM^ BLX UVB, Cedex, France). UVB radiation source was a fluorescent lamp that emitted an energy peak at 312 nm. After removed PBS, serum-free DMEM was added into cells and were incubated for 24 h. Cell viability was evaluated by MTT assay.

### 2.9. Wound Healing Assay

The wound-healing assay was used to determine HaCaT cell migration. The HaCaT cells were seeded at a density of 1 × 10^5^ cells/mL on a 24-well plate and incubated overnight at 37 °C in a humidified atmosphere of 5% CO_2_. The cells were then pre-treated for 24 h with samples at a concentration 100 μM. After that, the cell monolayer was formed, and the cells were scratched with a micropipette tip to create a wound gap. The cells were then washed with PBS and replaced by 100 μL of PBS. UVB-irradiation was irradiated to HaCaT cells at a dose 30 mJ/cm^2^. After removing PBS, the growth medium containing 100 μM of samples was added to cells. The cells were photographed at 0 and 24 h using inverted microscope (Leica DM IL LED, ×10 magnification). The wounding closure of HaCaT keratinocytes at 0 and 24 h after UVB-irradiation was analyzed using ImageJ. The wound area of cells was calculated based on the formula as below:(1)$$\mathrm{Wounding}\;\mathrm{closure}\;\left(\%\right)=\frac{A_{t0}-{\;A}_{t24}}{A_{t0}}\;\times \;100$$
A_t0_ is wound area measured at 0 h after UVB-irradiation, A_t24_ is wound area measured at 24 h after UVB-irradiation.

### 2.10. Real-Time Quantitative PCR

HaCaT cells (1 × 10^5^ cells/mL) were pre-treated with **2**, **4**, and **11** (25, 50, and 100 μM) in 24-well plates for 24 h. UVB-irradiation was irradiated to HaCaT cells at dose 30 mJ/cm^2^ and then incubated 24 h. The RNeasy Mini kit (Qiagen, Hilden, Germany) was used to extract total RNA from cells according to the manufacturer’s instructions. ReverTra Ace qPCR RT Mster Mix with gDNA Remover (TOYOBO, Osaka, Japan) was used to synthesize cDNA from the extracted total RNA. Real-Time quantitative PCR was performed using the synthesized cDNA as a template with an AriaMX (Agilent Technologies, Santa Clara, CA, USA). Using the THUNDERBIRD SYBR qPCRMix (TOYOBO, Osaka, Japan) for the real-time qPCR reaction. The real-time qPCR reaction conditions were initial denaturation at 95 °C for 60 s. cDNA samples were amplified for 40 cycles (95 °C for 15 s and 60 °C for 60 s). The samples tested in three replicate and COX-2, and iNOS gene expression levels were normalized to the corresponding GADPH level. The primer sequences are shown in Table 1.

### 2.11. Statistical Analysis

All experiments were performed in triplicate (*n* = 3) and data are expressed as mean values and standard deviation. A *p*-value of * *p* < 0.01; ** *p* < 0.05 was considered statistically significance.

## 3. Results and Discussion

### 3.1. Identification of the Isolated Compounds

Eleven compounds were identified or proposed in AE as shown in Figure 1 and Figure 2 and Table 2, identifications of the peaks were based on the retention characteristics, UV, and the accurate mass values, as well as comparisons with the reference standard compounds and the literature. Among these compounds, gallic acid, protocatechuic acid, 4-hydroxybenzoic acid, epicatechin, luteolin-7-*O*-β-d-glucoside, luteolin, quercetin, dihydrokaempferol, apigenin, and kaempferol were confirmed by comparison with authentic standard.

The AE was separated by chromatographic techniques to effort known six compounds (Figure 1). On the basis of NMR, MS data, and comparing with those previously reported, the compounds were identified as protocatechuic acid (**2**) [11], 4-hydroxybenzoic acid (**3**) [12], cirsimaritin (**6**) [13], and dihydrokaempferol (**9**) [14]. The structure of luteolin-7-*O*-β-d-glucoside (**5**) [15] and apigenin (**10**) [16] were elucidated by comparing LC-ESI-MS, *m*/*z* 447.0941 and 269.0459 [M − H]^−^ and co-elution of standards, in HPLC, respectively.

### 3.2. Cytotoxicity of Isolated/Standard Compounds

The effect of isolated compounds on normal HaCaT cells viabilities were performed by using MTT assay. Cytotoxicity of isolated compounds from AE on HaCaT cells was measured by MTT after the treatment 24 h. The results showed that **7** and **10** (100 μM) have toxicity while all other compounds displayed no toxicity (Figure 3).

### 3.3. UVB-Induced HaCaT Cell Damage

Human keratinocyte cells were irradiated with 30, 50, and 100 mJ/cm^2^ UVB and cultured with DMEM serum-free medium for 24 h. The cell viability was measured by MTT assay. The cell viability of the 30, 50, and 100 mJ/cm^2^ UVB-irradiated was 79.9%, 55.4%, and 48.0%, respectively (Figure 4). The result indicated that 30 mJ/cm^2^ was a proper UVB irradiation dose for inducing decrease in cell survival of HaCaT cells. Thus, we decided to use 30 mJ/cm^2^ for further experiments.

### 3.4. Protective Effects of Isolated and/or Standard Compounds against UVB-Irradiation

We determined the UVB protection effects of isolated and/or standard compounds with different concentrations 25, 50, and 100 μM. As shown in Figure 5, the cell viability pre-treated with isolated and/or standard compounds compared with UVB-group indicated that pre-treatment with 25, 50, and 100 μM of **2**, **4**, and **11** for 24 h promoted cell viability. Based on these results, pre-treatment with 25, 50, and 100 μM of **2**, **4**, and **11** were used for RT-qPCR.

Flavonoids have a chemical structure that consist of a benzene ring (A) condensed with a six-member ring (C) that has a phenyl ring (B) as substituent in the 2-position (Figure 6A). Flavonoids are divided into six group based on the substitution pattern of ring C. In the classification, both the oxidation state of the heterocyclic ring and the position of ring B are relevant (Figure 6B).

Based on the previous studies, 5, 7-OH substituents on the A-ring are needed for flavonoids to have good anti-inflammatory activity [17]. The distinction between luteolin (**7**), apigenin (**10**), and kaempferol (**11**) is the position of the hydroxyl group on the C ring. At a concentration of 100 μM, kaempferol (**11**) outperforms luteolin (**7**), and apigenin (**10**) in protecting UVB-induced HaCaT cells. According to the results, kaempferol (**11**) and quercetin (**8**) both have 5,7-OH substituents on the A ring, with the number and position of hydroxyl groups on the B ring differing. Kaempferol (**11**) having only one hydroxyl group on the C ring have significantly increased their activity.

### 3.5. Protocatechuic Acid, Epicatechin, and Kaempferol Rescue UVB-Induced Migration Defect in HaCaT Cells

HaCaT cell migration is critical for skin wound healing. The effect of **2**, **4**, and **11** on keratinocyte migration was assessed using an in vitro HaCaT cell scratch assay. After wounding, at 0 and 24 h the wound area surrounded by the edges of the wound monolayer was measured and expressed. Wounds pre-treated with 100 μM of **2** and **4** had a narrow region than wounds that had not been pre-treated (Figure 7A). Additionally, wound area with of **2**, **4**, and **11** showed percentage differences after 24 h incubation (Figure 6B). The findings revealed that (-)-epicatechin, and protocatechuic acid helped to reverse the effects of UVB on wound healing.

### 3.6. Effect of Protocatechuic Acid, Epicatechin, and Kaempferol on UVB-Irradiated Inflammatory Marker in HaCaT Cells

COX-2 and iNOS are the remarkable inflammatory markers, and over expression of these markers inhibited HaCaT cells growth and differentiation. RT-qPCR was used to examine COX-2 and iNOS expression. UVB irradiation increased the expression of COX-2 and iNOS markers in HaCaT cells in the current investigation. Meanwhile, pre-treatment of 100 μM of **2**, **4**, and **11** show diminished expression of both COX-2 and iNOS in HaCaT cells. This was shown in Figure 8A,B. The results show downregulation of COX-2 and iNOS in UVB-irradiated human keratinocytes.

The UV percentage of sunlight is responsible for skin damage, which occurs as a result of delicate and long-term exposure. Acute UVB irradiation of human skin causes a variety of cellular and pathological changes, including cell-cycle arrest, DNA damage, antioxidant defense system fatigue, and inflammation. Natural substances have captivated significant thought as skin protection in this regard [18]. UVB irradiation arbitrated inflammatory in this investigation, while the isolated and/or standard chemicals from AE prevented this.

COX-2 expression caused by UVB is linked to skin redness, the creation of pro-inflammatory cytokines, and the penetration of inflammatory cells. UVB irradiation increased iNOS expression and generation of its reactive product in HaCaT cells [19], which were successfully protected by pre-treatment with 100 μM of **2**, **4**, and **11**. Under normal circumstances, NO plays an important role in maintaining the homeostasis of normal skin function; nevertheless, excess NO production caused by increased iNOS has been linked to skin diseases such as redness, dermatosis, and psoriasis [20].

## 4. Conclusions

In conclusion, we investigate the effect of isolated and/or standard compounds from AE on anti-phototoxicity in human keratinocytes (HaCaT cells). Protocatechuic acid (**2**), epicatechin (**4**), and kaempferol (**11**) from AE showed that the inhibition of UVB-induced inflammation, which includes the improved expression of COX-2 and iNOS gene. Moreover, protocatechuic acid (**2**) and epicatechin (**4**) could potently enhance cell migration during wound closure. We suggest that protocatechuic acid (**2**), epicatechin (**4**), and kaempferol (**11**) are potential compounds for treating photoaging.

## Figures and Tables

**Figure 1 molecules-27-00440-f001:**
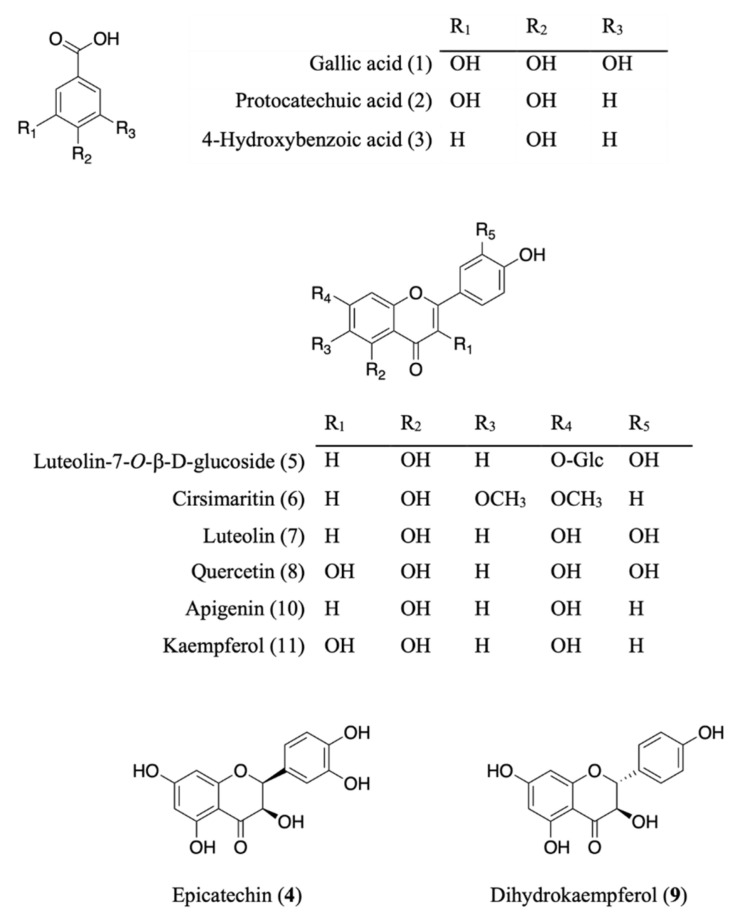
Chemical structures of isolated and/or standard compounds from AE.

**Figure 2 molecules-27-00440-f002:**
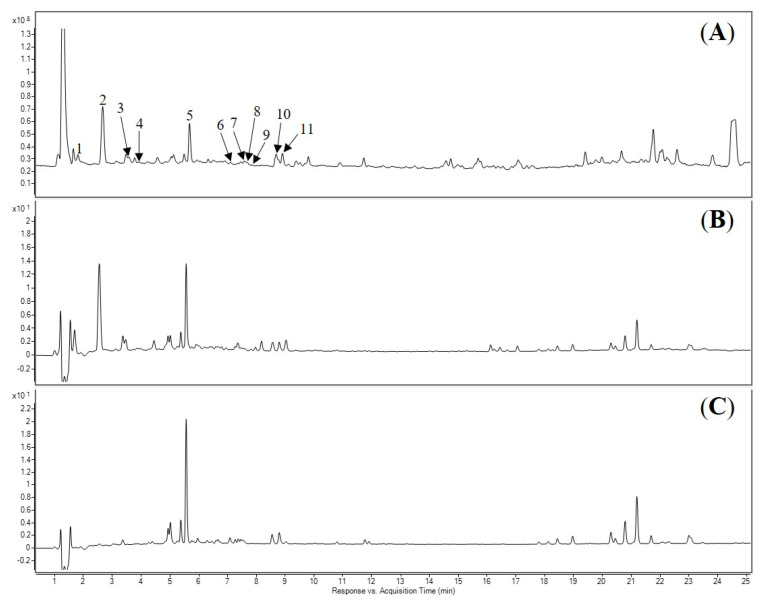
LC-DAD-MS chromatograms of the acetone extract of *E. phaseoloides* leaves. (**A**) Total ion chromatogram (TIC) spectrum. (**B**) Detected at 280 nm. (**C**) Detected at 330 nm. The numbering and identification of the peaks refer to Table 2.

**Figure 3 molecules-27-00440-f003:**
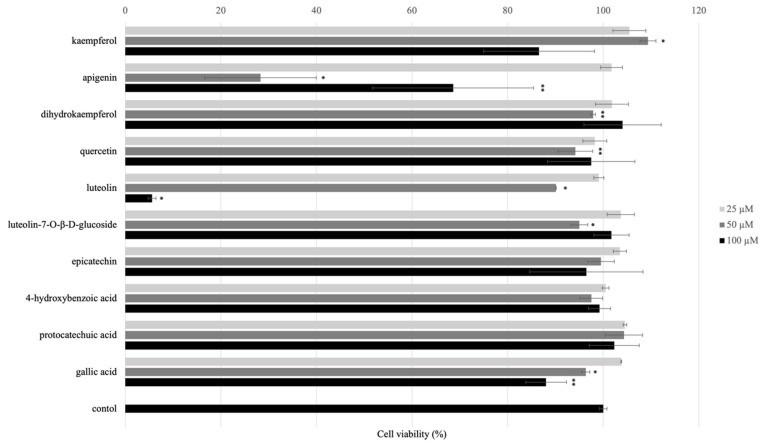
Effect of isolated and/or standard compounds on viability of the HaCaT cells. Cell viability was measured by MTT assay. HaCaT cells (1 × 10^5^ cell/well) were seeded to 96-well plate and then incubated overnight. The cell viability was performed after treated with isolated compounds for 24 h. Values are presented as the mean ± SD of three wells; * *p* < 0.01 compared with the control group; ** *p* < 0.05 compared with the control group.

**Figure 4 molecules-27-00440-f004:**
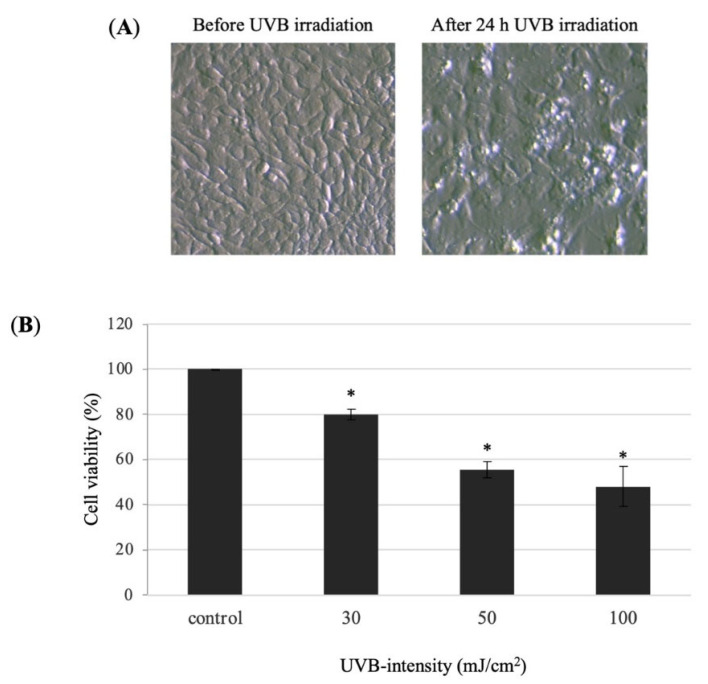
Effect of UVB radiation on the cell viability of HaCaT cells. (**A**) Morphology of HaCaT cells before and after 24 h UVB-irradiation (30 mJ/cm^2^). (**B**) The cell viability in the UVB-irradiated HaCaT cells was performed using MTT assay. HaCaT cells (1 × 10^5^ cell/well) were seeded into 96-well plate and then incubated for 24 h. The HaCaT cells were irradiated with 30, 50, and 100 mJ/cm^2^ of UVB. After UVB-irradiation, the HaCaT cells were incubated with DMEM serum-free medium for 24 h. Values are presented as the mean ± SD of three wells.; * *p* < 0.01 compared with the control group.

**Figure 5 molecules-27-00440-f005:**
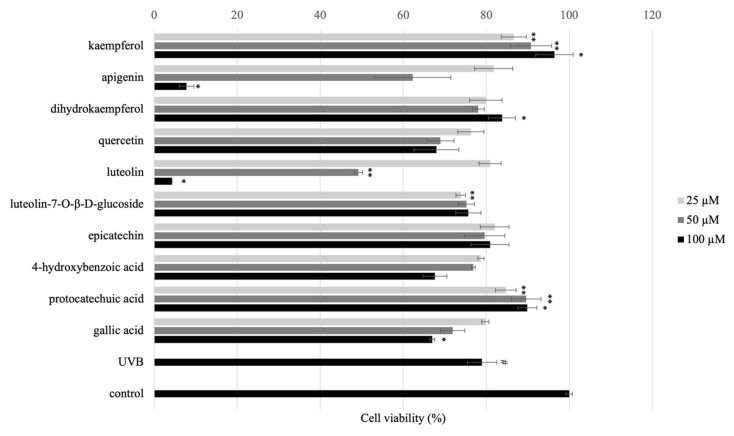
Phototoxicity of the pre-treatment of isolated and/or standard compounds in HaCaT cells (1 × 10^5^ cells/mL) with concentrations 25, 50, and 100 μM. After 24 h, UVB (30 mJ/cm^2^) was irradiated, and the cells were cultured in serum-free medium for 24 h. The cell viability was performed by MTT assay. The data are expressed as mean ± SD of at least three independent experiments in each group; # *p* ≤ 0.01, compared with the non-treated control; * *p* < 0.01; ** *p* < 0.05 compared with the UV control group.

**Figure 6 molecules-27-00440-f006:**
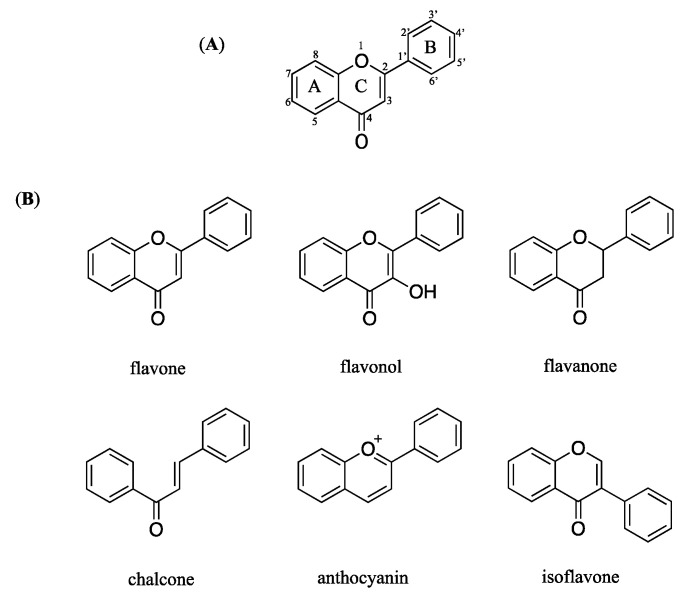
Chemical structure of flavonoid (**A**) general chemical structure (**B**) main structures classed of flavonoids.

**Figure 7 molecules-27-00440-f007:**
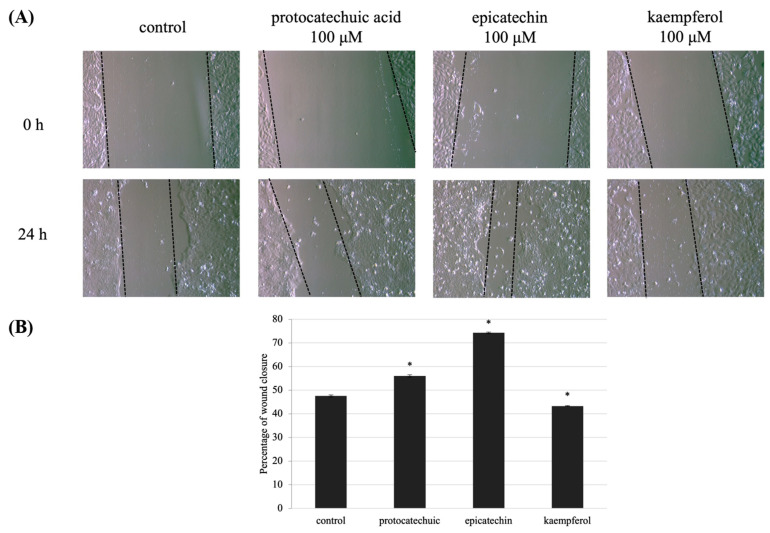
UVB-induced defect in HaCaT cells migration is decreased by protocatechuic acid (**2**), epicatechin (**4**), and kaempferol (**11**) pre-treatment. (**A**) Wound areas were formed in confluent layers of untreated and treated HaCaT cells prior to UVB irradiation. Phase-contrast images of the wounds were captured 0 and 24 h following irradiation. (**B**) The percentage of cell migration indicates the wound area at 0 and 24 h after making wound area and UVB-induced, compared with the initial area at 0 h. Values are presented as the mean ± SD of three wells.; * *p* < 0.01 compared with the control group.

**Figure 8 molecules-27-00440-f008:**
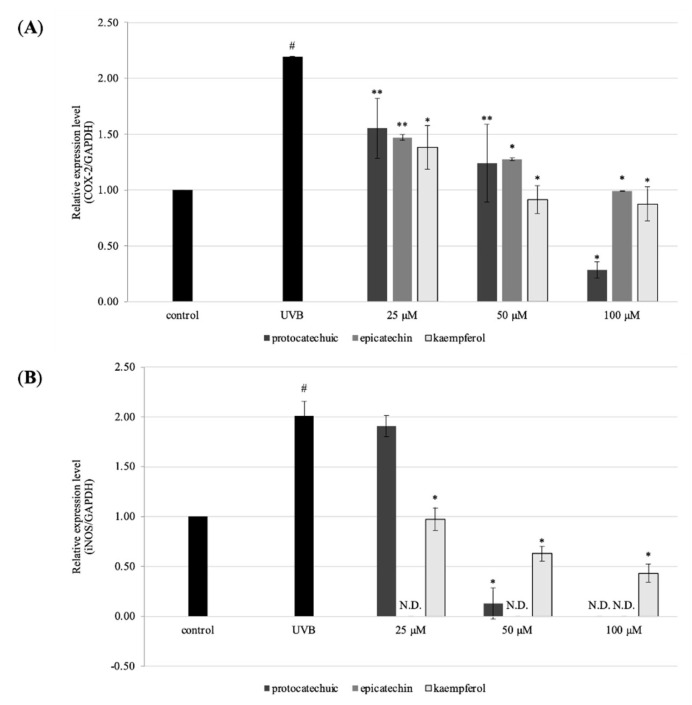
Effects of protocatechuic acid (**2**), epicatechin (**4**), and kaempferol (**11**) on the gene expression of HaCaT cells. Cells were pre-treated with 25, 50, and 100 μM for 24 h. After that, UVB (30 mJ/cm^2^) were exposed to monolayer HaCaT cells. After removed PBS, serum-free DMEM was added into cells and were incubated for 24 h. The gene expression was measured by RT-qPCR. (**A**) the gene expression level of COX-2, and (**B**) iNOS. The data are expressed as mean ± SD of at least three independent experiments in each group; # *p* < 0.01, compared with the non-treated control; * *p* < 0.01; ** *p* < 0.05 compared with the UV control group.

**Table 1 molecules-27-00440-t001:** PCR primer sequences used in this study.

Target	Sequence	
	Forward	Reverse
GADPH	5′-GCACCGTCAAGGCTGAGAAC-3′	5′-ATGGTGGTGAAGACGCCAGT-3′
COX-2	5′-AAGTTGGCAGCAAATTGAGCA-3′	5′-TCCTTTTCTCCTGTGAAGGCG-3′
iNOS	5′-TACTCCACCAACAATGGCAA-3′	5′-ATAGCGGATGAGCTGAGCAT-3′

**Table 2 molecules-27-00440-t002:** Characterization of the isolated and/or standard compounds from AE by LC-DAD-MS/qTOF.

Peak No.	Retention Time, t_R_ (min)	Formula	[M-H]^−^ (*m*/*z* exp)	Extract Mass	Error (ppm)	Identification
1	1.80	C_7_H_6_O_5_	169.0142	170.0215	−0.19	Gallic acid
2	2.67	C_7_H_6_O_4_	153.0199	154.0271	4.04	Protocatechuic acid
3	3.57	C_7_H_6_O_3_	137.0243	138.0322	−0.9	4-Hydroxybenzoic acid
4	3.90	C_15_H_14_O_6_	289.0734	290.0795	5.29	Epicatechin
5	5.69	C_21_H_20_O_11_	447.0941	448.1011	1.71	Luteolin-7-*O*-β-d-glucoside
6	7.27	C_17_H_14_O_6_	313.0716	314.0790	−0.26	Cirsimaritin
7	7.57	C_15_H_10_O_6_	285.0408	286.0477	1.24	Luteolin
8	7.65	C_15_H_10_O_7_	301.0358	302.0432	0.68	Quercetin
9	7.92	C_15_H_12_O_6_	287.0567	288.2500	−1.55	Dihydrokaempferol
10	8.65	C_15_H_10_O_5_	269.0459	270.0533	0.78	Apigenin
11	8.90	C_15_H_10_O_6_	285.0408	286.0477	1.24	Kaempferol

## Data Availability

Not applicable.

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
