# Peer review of "Anti-Phototoxicity Effect of Phenolic Compounds from Acetone Extract of Entada phaseoloides Leaves via Activation of COX-2 and iNOS in Human Epidermal Keratinocytes"

_molecules, 2022, doi:10.3390/molecules27020440_

Round 1

Reviewer 1 Report

The authors have addressed my concerns and the manuscript is now suitable for publication

Author Response

Thank you very much for your accepted. 

Reviewer 2 Report

Dear Authors, the manuscript is accepted as it is, but the authors should write the spectroscopic identification or at least they add references of the identified compounds.

Author Response

Thank you for your comments.

Based on your suggestions, the reference for each identified compounds has been added as shown in the revised manuscript. 

This manuscript is a resubmission of an earlier submission. The following is a list of the peer review reports and author responses from that submission.

Round 1

Reviewer 1 Report

The  manuscript including the uses of  Entada phaseoloides acetone extract as anti-phototoxic agents and the authors identify 11 compounds and they test three compounds for their phototoxic properties.

There is some comments and observations should be done:

  1. The title is long, so the authors should delete  (HaCaT Cells)
  2. In the Abstract part: epicatechin is mentioned as compound 5 but in all manuscript is mentioned as 4
  3. Why the authors choose the acetone extract for investigation, although they did not isolate nonpolar compounds like terpenes and fatty acids
  4. what is the meaning of 4-hydro-benzoic acid????
  5. The authors mentioned that they isolate  six phenolic compounds, protocatechuic acid (2), 4-hydro-benzoic acid (3), epicatechin (5), cirsimaritin (6), dihydrokaempferol (9), and apigenin (10) by various chromatographic techniques, and they did not mention Kaempferol, I would like to ask how they get the kaempferol for biological evaluation.
  6. The authors should add references for the  identification of isolated compounds
  7. I think these references will benefit for him:   Sobeh, M., El-Raey, M., Rezq, S., Abdelfattah, M. A., Petruk, G., Osman, S., ... & Wink, M. (2019). Chemical profiling of secondary metabolites of Eugenia uniflora and their antioxidant, anti-inflammatory, pain killing and anti-diabetic activities: A comprehensive approach. Journal of ethnopharmacology, 240, 111939.

       Sobeh, M., Petruk, G., Osman, S., El Raey, M. A., Imbimbo, P., Monti, D. M., & Wink, M. (2019). Isolation of myricitrin and 3, 5-di-O-methyl gossypetin from Syzygium samarangense and evaluation of their involvement in protecting keratinocytes against oxidative stress via activation of the Nrf-2 pathway. Molecules, 24(9), 1839.

8. some comments are highlighted in the manuscript.

Author Response

Thank you for kind suggestions in regarding to our manuscript. The following are our explanations/feedbacks for some of the suggestions. 

3. Why the authors choose the acetone extract for investigation, although they did not isolate nonpolar compounds like terpenes and fatty acids

Based on HPLC chromatogram DAD detector, the components of acetone extract are flavonoids. Thus, we focused on isolating flavonoid compounds.

5. The authors mentioned that they isolate  six phenolic compounds, protocatechuic acid (2), 4-hydro-benzoic acid (3), epicatechin (5), cirsimaritin (6), dihydrokaempferol (9), and apigenin (10) by various chromatographic techniques, and they did not mention Kaempferol, I would like to ask how they get the kaempferol for biological evaluation.

Six compounds, protocatechuic acid (2), 4-hydroxybenzoic acid (3), luteolin-7-O-glucoside (5), cirsimaritin (6), dihydrokaempferol (9), and apigenin (10) were isolated by various chromatographic techniques. However, kaempferol and epicatechin were proposed by LC-DAD-MS/qTOF. Thus, we evaluate their biological activity as well.

6. The authors should add references for the identification of isolated compounds

Thank you for your suggestion. We put references for all compounds the reference number 11-14 for isolated compounds.

The changes are highlighted in red in the manuscript.

Reviewer 2 Report

Mittraphab et al, show that the acetone extracts of entada phaseoloide have potential photo-protective effects in the skin. The manuscript is well written and the conclusion drawn are supported by the experimental data. There are a few issue to address.

If possible it would be extremely informative to repeat a few of the key experiments using primary skin keratinocytes. This would strengthen the potential therapeutic benefit of these compounds.

In Figure 7 Was the scratch assay experiment in figure 7 repeated more than once? As there are no error bars of stat for the graphs provided.

Analysis of caspase 3 transcript levels is not an appropriate measurement  of apoptosis. The authors should assess cleaved caspase 3 protein levels.

Author Response

Thank you for your kind suggestions in regarding to our manuscript. The following are our explanations/feedback to each of the comment. 

In Figure 7 Was the scratch assay experiment in figure 7 repeated more than once? As there are no error bars of stat for the graphs provided.

- According to migration assay, we repeated the experiment three times. In Figure 7, we modified the formula used for calculations and added error bars to the graph.

Analysis of caspase 3 transcript levels is not an appropriate measurement  of apoptosis. The authors should assess cleaved caspase 3 protein levels.

- Thank you for your suggestion. In further experiments, we will do Western blot for measure apoptosis cells.

Round 2

Reviewer 2 Report

The revised manuscript submitted by Mittraphab et al does not address my key concern with the study, the analysis of caspase 3. In the title and in the results sections the authors discuss that the compounds modulate caspase 3 activation and there evidence for this is changes in caspase 3 transcript levels. The authors cannot infer changes in caspase 3 activation from transcript levels. To analysis changes in caspase 3 activation the authors would have to look at cleavage of caspase 3 via western blot. Without this key evidence the authors would have to remove the claim in the title and text that caspase 3 activation was changed